# WASUP: Interpretable Classification with Weight-Input Alignment and Class-Discriminative SUPports Vectors

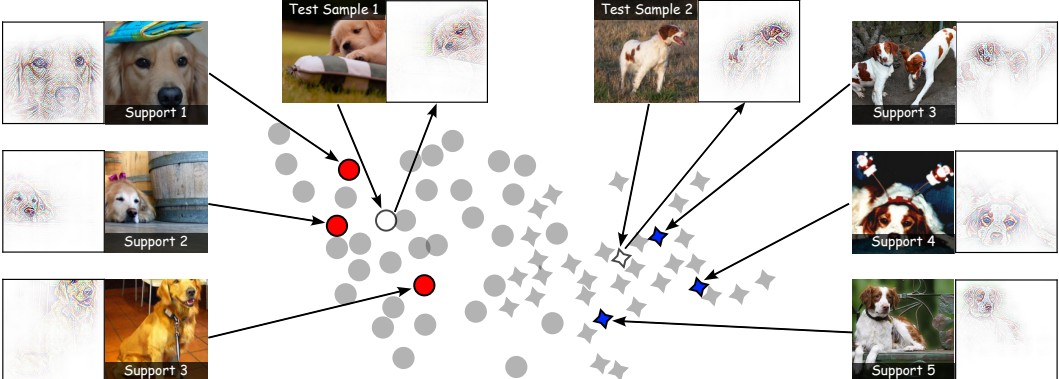

Figure 1: WASUP trains a neural network to extract class-representative support feature vectors (red and blue) from support images and classifies via computation of their similarities to the test images' latent vectors (white). Its B-cos architecture permits the computation of faithful local and global explanations. Stars and circles denote samples of different classes.

## Abstract

The deployment of deep learning models in critical domains necessitates a balance between high accuracy and interpretability. We introduce WASUP, an inherently interpretable neural network that provides local and global explanations of its decision-making process. We prove that these explanations are faithful by fulfilling established axioms for explanations. Leveraging the concept of case-based reasoning, WASUP extracts class-representative support vectors from training images, ensuring they capture relevant features while suppressing irrelevant ones. Classification decisions are made by calculating and aggregating similarity scores between these support vectors and the input's latent feature vector. We employ B-Cos transformations, which align model weights with inputs to enable faithful mappings of latent features back to the input space, facilitating local explanations in addition to global explanations of case-based reasoning. We evaluate WASUP on three tasks: fine-grained classification on Stanford Dogs, multi-label classification on Pascal VOC, and pathology detection on the RSNA dataset. Results indicate that WASUP not only achieves competitive accuracy compared to state-of-the-art black-box models but also offers insightful explanations verified through theoretical analysis. Our findings underscore WASUP's potential for applications where understanding model decisions is as critical as the decisions themselves.

## 1 Introduction

Deep learning is increasingly becoming part of everyday life, e.g., targeted advertisements, gaming, and high-stakes areas like banking and medicine. In the latter, identifying errors in the model's decision-making process is crucial to detect failure cases, which can be addressed by adding transparency to model prediction using explainable AI (XAI). Explanations are typically produced post-

hoc for black-box models using approximation methods, which estimate the contribution of each input feature to the prediction (Bach et al., 2015; Ribeiro et al., 2016; Selvaraju et al., 2017; Lundberg & Lee, 2017; Sundararajan et al., 2017; Petsiuk, 2018). While these methods can accurately compute feature contributions for low-dimensional inputs Lundberg & Lee (2017), they introduce approximation errors for high-dimensional inputs such as images (Adebayo et al., 2018).

As an alternative, *inherently interpretable models* provide explanations by design. For instance, logistic regression can be considered an inherently interpretable model since evaluating its coefficients directly explains the effect of each input feature towards the prediction. Translating these properties to complex functions implemented by neural networks was further fueled by the argument that if a technique is able to summarize the decision-making of a black box without any error, the black box could be replaced by an inherently interpretable model in the first place (Rudin, 2019). As a result, a growing research community is dedicated to developing inherently interpretable models, e.g., for decision-critical tasks like medicine Barnett et al. (2021); Kim et al. (2021); Wolf et al. (2023).

The explanations provided by these models can be categorized into local and global explanations, and evaluated in terms of axioms to verify if they are faithful Sundararajan et al. (2017) [1]. Local explanations focus on individual predictions, revealing how the model arrived at a specific outcome for a particular instance. In contrast, global explanations provide insight into the model's overall behavior. In logistic regression, global explanations are derived from the model's coefficients, which describe how each feature influences the prediction across all instances. To produce local explanations, a sample's specific feature values are multiplied by their corresponding coefficients, illustrating how each feature contributed to the particular prediction.

As outlined in Tab. 1, *locally* interpretable deep learning models include the work by Wang & Sabuncu (2022), who trained a non-parametric classification head with metric learning to classify an image by summing the softmax-normalized similarity scores between training image latent feature vectors and the test image latent feature vector. They proposed to provide local explanations by showing the most influential images alongside their corresponding scores for a given test sample, similar to the k-nearest-neighbor algorithm. While intuitive, the approach lacks contribution maps in pixel space to identify important input features. To this end, BagNets (Brendel & Bethge, 2019) classify local patches of an image individually and sum the probabilities of all patches for the final classification. Providing the class-probability maps of the individual patches' predictions as explanations shows which local structures showed evidence of a respective class. However, they lack faithful fine-grained attributions computed on a pixel-level, which are provided by B-cos networks (Böhle et al., 2022; Böhle et al., 2024). In these models, each forward pass is summarized as a linear transformation, and its resulting weight matrix then serves to compute local explanations in image space. More recently, they were used to guide knowledge distillation by Parchami-Araghi et al. (2024), who showed that a student network trained with knowledge distillation detects the same features as the teacher network, which explanation-guiding helped to overcome. However, B-cos models lack global interpretability because the weight matrix is dynamically computed for each input.

Table 1: Comparison of inherently interpretable models and their properties.

| Model | Local Expl. | Global Expl. | Faithful Expl. |
|---|---|---|---|
| NW-Head Wang & Sabuncu (2022) | ✓ | ✓ | ✗ |
| ProtoPNet Chen et al. (2019) | ✓ | ✓ | ✗ |
| BagNets Brendel & Bethge (2019) | ✓ | ✗ | ✓ |
| B-Cos Böhle et al. (2022) | ✓ | ✗ | ✓ |
| WASUP (ours) | ✓ | ✓ | ✓ |

Models that provide global interpretability include concept bottleneck models (Koh et al., 2020), which train an encoder to predict concepts present in an image and a classifier to predict the target class from these concepts, which serve as local explanations during a prediction. However, they require access to concepts, i.e., meta information on each concept present in an image, e.g., the

---

[1] Throughout this manuscript, we define faithful explanations as explanations that satisfy all of the axioms proposed by Sundararajan et al. (2017): Completeness, Sensitivity, Implementation Invariance, Dummy, Linearity, and Symmetry-Preserving.

color of the tail, legs, or head of a bird. While the concepts themselves provide global and local explanations and are manually pre-defined, concept bottleneck models lack explanations in the image domain like the Nadaraya-Watson head (Wang & Sabuncu, 2022). The most prominent inherently interpretable model is ProtoPNet (Chen et al., 2019) and its extensions (), which provide local and global explanations. Its key idea is to learn class-specific prototypical parts across the training set to facilitate reasoning as "this part of an unseen test image looks like that part from my training images." ProtoPNet implements its reasoning by computing similarity scores between learned latent feature vectors and (test sample) latent feature vectors. To provide image space explanations, it upsamples similarity maps from latent space into the input domain. However, Hoffmann et al. (2021) found that the explanations provided by ProtoPNet can be misleading and called for more rigorous evaluations. Sacha et al. (2024) proposed a benchmark to evaluate spatial misalignment of its prototypical parts, and (Wolf et al., 2024) showed that the explanations are generally not faithful, primarily due to an absence of a spatial relationship between the input space and latent space in deep neural networks, which generalizes to other prototypical networks like Pip-Net (Nauta et al., 2023), ProtoTrees (Nauta et al., 2021), XProtoNet (Kim et al., 2021), or the Deformable ProtoPNet (Donnelly et al., 2022). While prototypical part-based case-based reasoning has shown to be the most human comprehensible explanation (Nguyen et al., 2021; Kim et al., 2023; Jeyakumar et al., 2020), the number of explanations to be evaluated in its current implementations grows with the number of prototypical parts per class learned by the model.

We propose WASUP, an inherently interpretable neural network that fulfills all three properties: local, global, and faithful explanations (see Tab. 1). We inject case-based reasoning into WASUP's decision-making to provide local and global explanations that reduce the complexity of explanations to be evaluated. WASUP can implement any neural network architecture (e.g., ResNets He et al. (2016), Transformers Vaswani (2017), Mamba Gu & Dao (2023)) in its feature extractor without limiting their computational capacity. Specifically, the model extracts support vectors from training images that capture only class-representative features by suppressing negative contributions from other classes. Classification is performed by computing and summing similarity scores between these support vectors and the latent feature vector of the input image (see Fig. 1). Levering B-cos networks ensures that both the support vectors and their similarity to test images are directly and faithfully explainable in the input space, enabling an intuitive understanding of the model's reasoning. In summary, the support samples serve two purposes: (i) they help the model capture the high variability between different classes in the data, and (ii) they provide the user with examples that illustrate which parts of the class-representative training images the feature extractor focuses on. Our contributions are as follows:

- We propose WASUP, an inherently interpretable network that provides faithful local and global explanations for image classification with either single or multiple labels.

- We prove that explanations provided by WASUP fulfill the axioms required to be faithful.

- We empirically evaluate WASUP and its explanations on three tasks (fine-grained single-label image classification on Stanford Dogs, pathology prediction on RSNA, multi-label image classification on Pascal VOC), and three architectures with different numbers of parameters (DenseNet121: 8M, ResNet50: 26M, a hybrid vision transformer: 81M).

- We demonstrate how the explanations provided by WASUP enable model debugging.

## 2 BACKGROUND

**Learning Class-Discriminative support vectors** Wang & Sabuncu (2022) proposed to classify images with the non-parametric Nadaraya-Watson head and a convolutional neural network $\mathcal{F}_\theta$ with parameters $\theta$ extracting latent vector of dimension $d$. During training, they randomly sample a number of support vectors $v_k^c$ for every batch per class from the training set $\{(\mathcal{I}_k, y_k)\}_{k=1}^N$, $V^c = \{v_k^c = \mathcal{F}_\theta(\mathcal{I}_k) \mid y_k = c\}$, with $\Omega(V^c) = N_s$. This selection is executed for every training batch, thereby directing the optimization of the feature extractor $\mathcal{F}_\theta$ to form distinct, class-specific clusters within the latent feature space.

For classification, a test image $\mathcal{I}$ is first transformed into its latent representation $\mathcal{F}_\theta(\mathcal{I})$. The probability that $\mathcal{I}$ belongs to class $c$ is computed by applying the softmax function to the negative squared

L2 distances between $\mathcal{F}_\theta(\mathcal{I})$ and each support vector $v_k^c$:

$$p(y = c \mid \mathcal{I}) = \sum_k \frac{\exp\left(-\|\mathcal{F}_\theta(\mathcal{I}) - v_k^c\|_2^2\right)}{\sum_{k'} \exp\left(-\|\mathcal{F}_\theta(\mathcal{I}) - v_{k'}^{c'}\|_2^2\right)}.$$

This classification strategy leverages the learned latent space structure to assign class probabilities based on proximity to the nearest support clusters.

Local explanations show the images of the most influential support samples in terms of softmax probabilities (see Fig. 2). At first glance, these explanations seem intuitive, but the main limitations are two-fold: (i) Computing probabilities from (inverted) Euclidean distances with softmax: Regardless of the input (e.g., in an adversarial attack), the model explanations will suggest that at least one support sample is similar with sufficient probability. Additionally, the softmax probabilities are inapplicable to multi-label classification tasks out of the box since they sum to one. Thus, the model cannot predict three classes simultaneously under a standard decision threshold of 0.5. (ii) No pixel-level explanations: While explanations using similar input images are intuitive for humans, the lack of attribution maps hinders a further understanding of the network's mechanisms and the detection of biased predictions.

**Faithful Explanations with B-cos Networks**  B-cos networks proposed by Böhle et al. (2022) impose weight-input alignment between the layer weights and input by re-formulating the scalar product used by neural network computations, removing biases and standard non-linearities. This forces the network to focus on the most salient features across the training set and facilitates the summary of each forward pass as an input-dependent linear equation. Each input feature's contribution can directly be traced by its corresponding weight in the corresponding transformation matrix.

Mathematically, the B-cos transformation computes a modified dot-product, that increases weight-input alignment its exponent $B$, between an input vector $x$ and a layer weight $w$:

$$\text{B-cos}(x; w) = \|\hat{w}\|\|x\||\cos(x, \hat{w})|^B \times \text{sgn}(\cos(x, \hat{w})), \tag{1}$$

with $\hat{w} = w/\|w\| \implies \|w\| = 1$, cos the cosine, sgn the sign function, and $\times$ the real valued multiplication. While it is bounded by the magnitdude of $\|x\|$, it effectively computes an input-dependent linear equation, in matrix form expressed by:

$$\text{B-cos}(x, \mathbf{W}) = \tilde{\mathbf{W}}(x)x, \qquad \text{with} \qquad \tilde{\mathbf{W}}(x) = |cos(x, \hat{\mathbf{W}})|^{B-1} \odot \hat{\mathbf{W}}, \tag{2}$$

where $|cos(x, \hat{\mathbf{W}})|^{B-1}$ scales the elements of the fixed weight matrix $\tilde{\mathbf{W}}$ ($\odot$ the elementwise multiplication). A neural network $\mathcal{F}_\theta$ with parameters $\theta$ that computes a series of these equations is called piece-wise linear and its forward pass is summarized by the single input-dependent weight matrix:

$$\mathcal{F}_\theta(x) = \tilde{\mathbf{W}}_L(a_L)\tilde{\mathbf{W}}_{L-1}(a_{L-1})\ldots\tilde{\mathbf{W}}_1(a_1 = x)x = \left(\prod_{j=1}^L \tilde{\mathbf{W}}_j(a_j)\right)x = \mathbf{W}_{1 \to L}(x)x. \tag{3}$$

Thus, the contribution maps in terms of a pixel location $(m, n)$ in image space for a single forward pass are faithfully computed with:

$$\phi_j^l(x)_{(m,n)} = \sum_{ch}\left[[\mathbf{W}_{1 \to l}]_j^T \odot x\right]_{(ch,m,n)}, \tag{4}$$

with $l$ any layer and $j$ the index of the row (neuron) in $\mathbf{W}_{1 \to l}$ (note that this allows to explain any intermediate layer's $l$ neurons $j$). Since B-cos networks are trained with binary cross-entropy loss, features negatively attributing the class logit cannot increase its probability and are not visualized.

Böhle et al. (2022) propose to compute RGBA explanations by normalizing each color channel [ch, 1-ch][2] to sum to 1 to maintain the angle of each color pair and to then scale RGBA values into the range [0..1]. Finally, the alpha value (pixel opacity) is computed with the 99.9th percentile $p_{99.9}$ of the resulting weight $w$ as $\min\left(\left[\|w_{(m,n)}\|_2\right] / \left[p_{99.9}(\|w_{(m,n)}\|_2)\right], 1\right)$.

---

[2]B-cos networks require an image encoding of [R, G, B, 1-R, 1-G, 1-B] in the channel dimension to uniquely encode colors and to mitigate favoring of brighter regions.

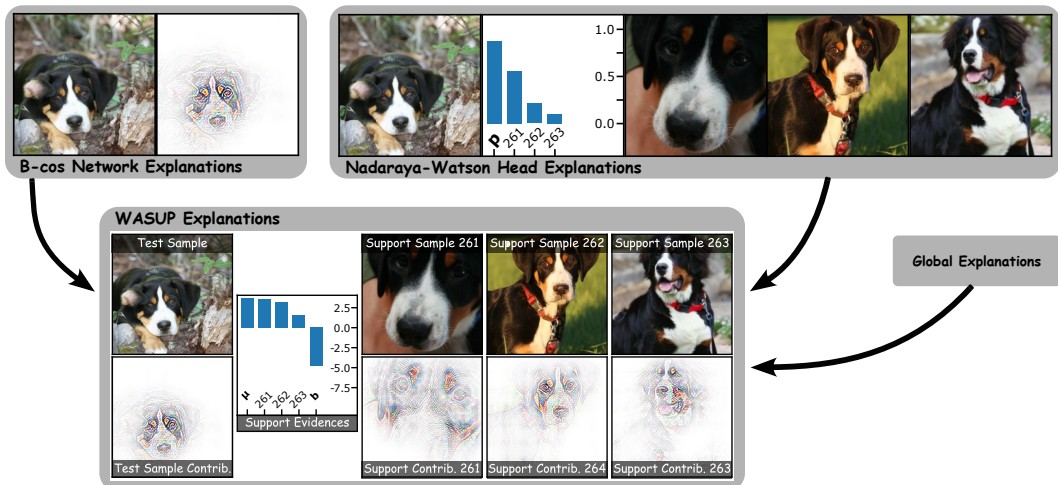

Figure 2: While the Nadaraya-Watson head Wang & Sabuncu (2022) provides explanations following case-based reasoning, and B-cos networks Böhle et al. (2022) faithfully explain the image transformation of a neural network, WASUP combines both properties, thereby adding global explanations by extracting a fixed set of class-representative, explainable support vectors.

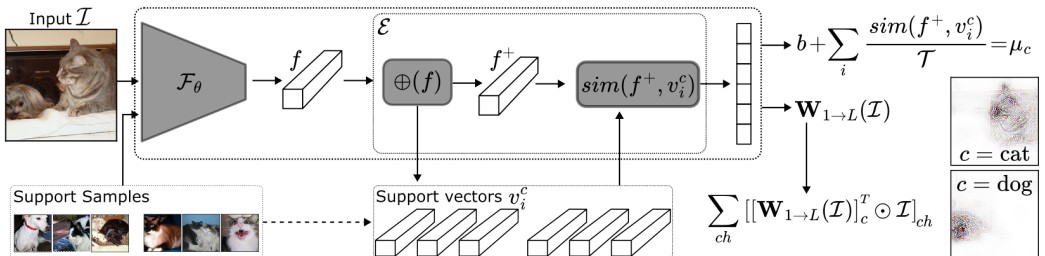

Figure 3: WASUP consists of a feature extractor $\mathcal{F}$ with model weights $\theta$ that extracts latent vectors $f$. The evidence predictor $\mathcal{E}$ converts them into positive-valued vectors $f^+$, which, given other images from the training set, can also serve as support vectors $v_i^c$. Then, it computes the similarity between the input image's latent vector $f$ and support vectors $v_i^c$. Summing the class-specific temperature-normalized similarity scores and adding a bias $b$ yields the class logit $\mu_c$. The B-cos transform facilitates the summary of a forward pass as a single weight-matrix $\mathbf{W}_{1 \to L}(\mathcal{I})$, which is leveraged to compute faithful contribution maps.

The obvious strength of B-cos models is that they are able to extract local explanations of a single forward pass that resemble the transformation of an input image with the network with high fidelity, which we prove to fulfill the axioms required for faithful explanations in the appendix A.1. Nonetheless, they do not provide global explanations (see Fig. 2).

## 3 METHODS

We propose WASUP, which combines the strengths of B-cos networks and the Nadaraya-Watson head, to obtain an *inherently interpretable* neural network that implements *case-based reasoning* and provides *faithful local* and *global* explanations, thereby reducing the complexity of explanations to evaluate. WASUP achieves this by extracting class-representative support vectors from training images for similarity-based classification and by leveraging the B-cos transform to faithfully explain its decision, see Fig. 2.

## 3.1 ARCHITECTURE

WASUP consists of a feature extractor $\mathcal{F}_\theta$ (see Fig. 3) employs a B-cos backbone and transforms an image $\mathcal{I}$ into latent vectors $f = \mathcal{F}_\theta(\mathcal{I}) \in \mathbb{R}^d$, with $\mathcal{I} \in \mathbb{R}^{ch \times H \times W}$, with $d$ being the latent dimension, $ch$, $H$, and $W$ the image's channels, height, and width, respectively, and $\theta$ the model weights ($\mathcal{F}_\theta$ can be any B-cos neural network architecture).

The evidence predictor $\mathcal{E}$ consists of a function $\oplus : \mathbb{R}^d \to \mathbb{R}^d_{\geq 0}$ that transforms the real-valued vectors $f$ into non-negative vectors $f^+ = \oplus(f)$, and a similarity measure that computes similarities between support vectors $v_i^c \in F^c = \{f_k^+ = \mathcal{F}_\theta(\mathcal{I}_k) \mid y_k = c\}$ and $f^+$ as $sim(f^+, v_i^c) \in \mathbb{R}$, with $\Omega(F^c) = N_s$ and $i$ denoting the $i$-th support vector of class $c$. Logits are computed for class $c$ as the sum of its similarity scores:

$$\mu_c = b + \sum_{v_i^c} \frac{sim(f^+, v_i^c)}{\mathcal{T}}, \tag{5}$$

where $b$ represents a fixed bias term, $i = 1, \ldots, N_s$ indexes the support samples, and $\mathcal{T}$ is a temperature to improve convergence by scaling the logits' magnitude appropriately. Examples of suitable transformation functions $\oplus$ include the exponential function, absolute value, and ReLU.

## 3.2 TRAINING AND OPTIMIZATION

Following Wang & Sabuncu (2022), we randomly sample the training set to extract $N_s$ support vectors $v_k^c$ per class to encourage the model to form distinct, class-specific clusters in latent space. After every epoch, we compute sets of latent vectors $\hat{V}^c = \{f_k^+ = \mathcal{F}_\theta(\mathcal{I}_k) \mid y_k = c\}_{k=1}^N$, which we each cluster using k-means to extract centroids $\gamma_i^c$. Then, we replace each centroid $\gamma_i^c$ with the closest latent vector to yield support vectors from the same class, defined as:

$$v_i^c = \arg \min_{f_k^+ \mid y_k = c} \|f_k^+ - \gamma_i^c\|_2, \tag{6}$$

which facilitates case-based reasoning similar to k-nearest-neighbors, as only features descriptive of the class can increase the probability of that class by definition of WASUP.

We follow the argument of Böhle et al. (2024) that training with binary cross entropy loss increases the alignment pressure induced by B-cos networks, and compute the loss for every batch of $N_b$ samples and $C$ classes as:

$$\mathcal{L}_{BCE} = -\frac{1}{N_b} \sum_{\hat{b}=1}^{N_b} \sum_{c=1}^C \left[ y_{\hat{b},c} \log(\hat{y}_{\hat{b},c}) + (1 - y_{\hat{b},c}) \log(1 - \hat{y}_{\hat{b},c}) \right], \tag{7}$$

with $\hat{y}_{\hat{b},c} = \sigma(\mu_{\hat{b},c})$, $\sigma$ denoting the sigmoid function. If trained with cross-entropy loss, it could potentially converge towards optima in which activation of the wrong classes is still relatively large, and thus, the alignment pressure would fail.

## 3.3 EXPLAINABILITY

**Global Explanations** While global explanations provided by WASUP are ideal for a developer to debug a model, they also facilitate case-based reasoning, which is most intuitive to understand for humans (Nguyen et al., 2021; Kim et al., 2023; Jeyakumar et al., 2020).

We propose to first evaluate what is encoded in each support vector by computing the RGBA explanations and contribution maps in terms of the similarity measure to its input image: B-cos$(v_i^c, \hat{v}_i^c) = \|v_i^c\|$, since $\hat{v}_i^c = v_i^c / \|v_i^c\| = 1$ and $\cos(v_i^c, \hat{v}_i^c)^B = 1$ for any $B$.

We observed negative attributions for the global features during initial experiments. While only visualizing features with positive class contributions is reasonable during the classification of an input image, these negative attributions can serve as means of debugging a model if carefully evaluated. The definition B-cos$(v_i^c, \hat{v}_i^c) = \|v_i^c\|$ implies that negative attributions are a means of the model to downscale the range of the B-cos similarity (accordingly, we observed the negative attributions in the corners of the images). This indicates the temperature scaling in Eq. 5 was chosen too small, and re-training with a more significant temperature helped to overcome this issue.

Additionally, WASUPś decision-making is based on a fixed set of latent support vectors. Hence, we can compute the inter- and intra-class similarities to gain insights into the latent space learned by the feature extractor.

**Local Explanations**  Local explanations, shown in Fig. 2 are threefold: First, the temperature-scaled similarity scores $\frac{sim(f^+, v_i^c)}{\mathcal{T}}$ (support evidence) provided by the evidence predictor $\mathcal{E}$ quantify the alignment of a test image with a support sample, and their proportion of the log-probability. Second, since a test image is assigned to a class if it is similar to that class's support sample (see Fig. 2), understanding which parts of the support sample's image are compressed in the support vectors themselves is crucial, i.e., explaining the model mechanisms. The B-cos properties of WASUP allow us to faithfully compute the RGBA explanations in terms of the similarity measure. The prediction of our model is a linear transform with the temperature and the bias term. Thus, we compute contribution maps in terms of the class logit (test contribution) following the linearity axiom (Sundararajan et al., 2017) ($\phi = \alpha\phi_1 + \beta\phi_2$ iff $\mathcal{F}(x) = \alpha\mathcal{F}_1(x) + \beta\mathcal{F}_2(x)$). Therefore, the test sample contributions highlight the image features that aligned with the support vectors. Third, the global RGBA explanations (support contributions) of the support samples allow the user to check for the intersection of support contributions and test contributions, similar to k-nearest-neighbor.

**Theoretic Evaluation**  To thoroughly assess the interpretability of WASUP, we conduct a theoretical evaluation of its explanations based on the axioms defined by Sundararajan et al. (2017), namely *Completeness, Sensitivity, Implementation Invariance, Dummy, Linearity,* and *Symmetry-Preserving*. Our proofs (see App. A.1), confirm that WASUP's explanations adhere to these principles, thereby validating the model's capability to provide faithful and comprehensive insights into its decision-making processes.

## 4 EXPERIMENTS AND RESULTS

We evaluate WASUP on three tasks with three backbone architectures as feature extractors. The tasks are (i) multi-label image classification on Pascal VOC (Everingham et al., 2009), (ii) fine-grained image classification with mutually exclusive labels on Stanford Dogs (Aditya et al., 2011), (iii) pathology prediction with mutually exclusive labels on RSNA (Shih et al., 2019). The backbone architectures are B-cos versions of DenseNet121 (Huang et al., 2017), ResNet50 (He et al., 2016), and a Hybrid Vision Transformer (Xiao et al., 2021) (VitC), with $\sim$8M, $\sim$26M, and $\sim$81M trainable parameters, respectively.

Compared to black-box versions of the backbones (see Tab. A.2), WASUP performs on par on Pascal VOC ($\geq$ +0.29%), worse on Stanford Dogs ($-3.7\%$ Acc), and comparable on RSNA ($-0.65\%$ Acc); take into account that we train the model from scratch on RSNA.

**Pascal VOC (Everingham et al., 2009)**  Fig. 4 illustrates explanations of WASUP for multi-label image classification on Pascal VOC with a DenseNet121 backbone. As seen in Figs. 4 (a) and (b), WASUP shifts its focus in the test image depending on the class to predict, i.e., the corresponding column $c$ in $[\mathbf{W}_{1 \to L}]_c^T$: To predict the cat class, only the area around the cat is present in the contribution map. When predicting the human class, for which the model computed very high support evidence, the complementary area of the image is contained in the contribution map. The support contributions show that the model is particularly sensitive to the eyes of a cat. In the test sample in Fig. 4 (a) and (b), only the left half and one eye of the cat are visible, which explains the comparably low support evidences in this particular instance. Arguably, the support contributions of the human class in Fig. 4 (b) suggest that the model learned to look for human eyes in support sample 42, which shows high contributions from the two lights on the ceiling. For support samples 43 and 44, the whole heads of the humans are the primary contributions to the support vectors.

Observing the wrong prediction presented in Fig. 4 (c) for the class potted plant, we see that the support contribution map of support sample 45 focuses on the soil in the pot, while the support samples 46 and 47 focus on the actual plants themselves. We see in the contribution map of the test sample that the model mistakes the flower on the left-hand side and the artificial flowers on the right-hand side with this class. The model correctly classified the TV/monitor class for the same sample, presented in Fig 4 (d), in which the test contribution map primarily highlights the TV. Here,

the support contributions of sample 57 show that the support vector carries information on all three monitors present in the image. In the case of support sample 58, the support contribution shows high contributions from the keyboard in addition to the monitor, indicating that the model picked up this bias from the data. Support contribution 59 focuses on the TV instead of background objects.

We present additional results for other test images and backbones in Figs. A.5-A.7. In many test sample instances (e.g., compare Fig. A.5 (a) vs. (b), Fig. A.5 (c) vs. (d), Fig.A.7 (a) vs. (b)), we observe a strong shift in contributions depending on the class prediction. This demonstrates that the latent vectors of samples with multiple labels efficiently encode features for all classes that need to align with the class support vectors. In Fig. A.6 (a)-(d), we see that the model predicted four classes, for which, according to the dataset labels, only a single class was actually contained (chair). However, the explanations look very reasonable. Indeed, there is a table in the scene; whether it is a dining table is obviously debatable. There is also a child in the image, so predicting humans is not unreasonable. Lastly, the potted plant class logit is very low, and the test contribution map highlights the flower on the table.

For the test sample predicted in Fig. A.7 (d) and (e), we see the correct classification of humans with high support evidence scores (note how the model extracted the human for support sample 44 in very dark lighting conditions), and WASUP believed that the martial arts club logo on the person's outfit resembled the logo of a drink.

**Stanford Dogs (Aditya et al., 2011)** We evaluate WASUP with a ResNet50 backbone on the Stanford dogs dataset. The contribution maps for the test samples in Figs. A.8,A.9,A.10 show that WASUP consistently focuses on the dogs and discards background elements such as humans, especially in Figs. A.8 (a) and (b).

The wrong classification of the test sample presented in Fig. A.8 (c) shows that primarily the head of the dog contributed to an increased similarity with the support vector of support sample 46, whose support contribution focuses on the head, too. In contrast, the support contributions of support samples 45 and 47 both indicate that the support vectors contain large portions of the body of the dog. Evaluating the explanations of the correct class in Fig. A.8 (d) shows that the support sample 33 containing two dogs has high support evidence, as there are two dogs of its class in the test image. However, the low activation of support samples 34 and 35, which both contain one dog, resulted in little support evidence, indicating that the number of instances is encoded into support vectors.

The different magnitude in the support evidence suggests that the model benefits from increasing the number of support samples in settings with high intra-class variability, which manifests in the visualization of support samples and their contribution maps. Considering the support evidence of support sample 162 in Figs. A.10 (a) and (b), its support evidence for the test sample in Fig. A.10 (a) is lower compared to the test sample in Fig. A.10 (b), which only contains the lower portion of the head like support sample 162. The support contributions of the other support samples that focus on the eyes and the body show the inverse behavior of the two respective test samples. For some classes, we observe similar support evidence for all support samples, e.g. Fig. A.9 (a) and (b), suggesting lower intra-class variability within the training data for the particular classes.

**RSNA (Shih et al., 2019)** We utilize the RSNA dataset to showcase WASUP on a decision-critical task to differentiate between images with lung opacity and others (healthy and unhealthy without lung opacity, indicated by the presence of bounding boxes) from chest X-ray images. Here, we want to verify reasonable decision-making by evaluating explanations in terms of bounding boxes. An explanation of a classifier trained to predict lung opacity should focus on an area inside the bounding box, illustrated in Figs. A.11-A.12.

Here, the model has to learn very subtle intra- and inter-class differences. Evaluating the prototypes in Fig. A.13 shows only positive contributions as desired. In the case of the three non-occluded support samples on the left, we immediately see that the sample is likely an outlier in that it has some disease without opacity. The second and third support contributions are primarily highlighted within the lungs and bounding boxes for the respective classes. Interestingly, the model seems to exclude atypical objects from the healthy support vectors: Consider the third non-occluded support contribution that does not contain contributions for the dark clamp present in the support sample. However, the first two support contributions of each class contain most of the input and are, thus, hard to interpret.

The predictions in Figs. A.11 (a)-(d) exemplify how the support evidence varies across the class's support samples and suggest that the increasing number of support samples allowed WASUP to reproduce the class's latent space with a likely high variability. Additionally, we observe a high degree of intersection between significant test sample contributions and bounding boxes for the occluded class in Figs. A.11 (a)-(d) and that the focus of the model is within the lungs for non-occluded samples in Fig. A.12, indicating that the model indeed learned to differentiate occlusion from no-occlusion using medically reasonable regions.

**Debugging WASUP**   While the experiments primarily highlighted the explainability features from the view of a user, we now exemplify how the global and local explanations help to identify flaws in the optimization process. Revisiting the model trained on RSNA, we detail how to use the global explanations of WASUP to gain more insights into the model behavior. We observe that the support evidence of support sample 0 of the correct classifications in Figs. A.12 (a) and (b) is almost absent compared to the wrong classification in Fig. A.12 (c). Considering that support contribution 0 contains large parts of the input, this raises suspicion. In addition, we see that the only significant evidence for the correct class depicted in Fig. A.12 (d) comes from support sample 3, which suffers from contributions across the whole input as well. As a result, we evaluate the latent space learned by WASUP in Fig. A.14 and find that support vectors 0 and 3, although of different classes, are very close in the t-SNE projection. We attribute this to a weak latent representation, suggesting that the model has yet to learn a more class-separable latent space.

Next, we evaluate whether the model indeed learned class-representative support vectors (see Fig. A.15) by computing the Silhouette score (0.655) (Rousseeuw, 1987) and plotting the inter- and intra-class B-cos similarities of WASUP trained on Pascal VOC with a DenseNet121 backbone in Fig. A.16. We observe that the intra-class similarities on the diagonal are indeed higher than those off-diagonal, suggesting good class distinction in latent space. Additionally, the average distance between some classes is a bit higher, e.g., dining table, chair, and bottle. This is expected, as images containing a dining table often include a chair and a bottle, requiring the model to extract a latent vector capable of predicting all three classes with sufficient confidence.

## 5   DISCUSSION AND CONCLUSION

Current inherently interpretable models either lack global explanations, fail to explain the internal mechanisms of the deep learning model, or cannot provide explanations that faithfully explain their decision-making. To this end, we proposed WASUP for interpretable classification by providing faithful local and global explanations by leveraging case-based reasoning and weight-input alignment achieved with B-cos neural networks. Our theoretical analysis shows that our proposed method's explanations satisfy the required axioms for faithfulness. We evaluated WASUPon three tasks using three backbone architectures, demonstrating how the complex mechanisms learned by neural networks can be decomposed into intuitive explanations with limited complexity.

Additionally, we exemplified how its global and local explanation properties enable effective model debugging from a developer perspective while giving intuitive explanations of its decision-making. We focused on creating a model that yields comprehensible explanations, which can, if desired, decompose each test contribution into individual components based on the similarity measure. This generates additional contribution maps for each support sample, facilitating further validation in cases with high intra-class variability, albeit with an increase in complexity.

While k-means clustering is fast to optimize, WASUP introduces additional overhead by requiring the computation of support vectors, which is a potential limitation in cases with many classes. However, as demonstrated with the Stanford Dogs dataset comprising 120 classes, WASUP is readily applicable out of the box to datasets with hundreds of classes. Moreover, note that the number of classes in decision-critical fields such as medicine is generally smaller. In future work, we are curious to extend WASUP by conformal prediction to automatically show the explanations of a set of possible classes as opposed to the most likely one in single-label tasks.

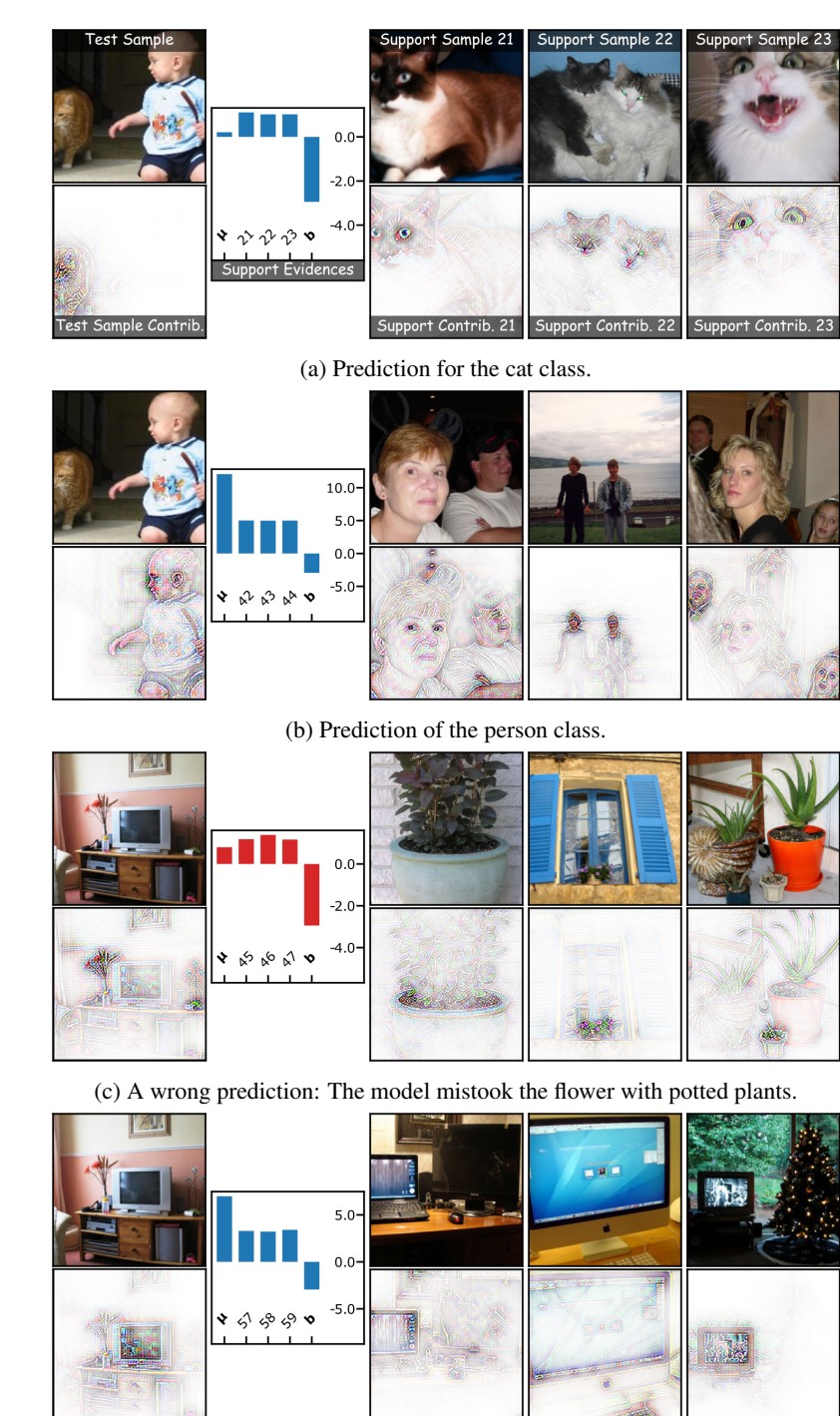

(a) Prediction for the cat class.

(b) Prediction of the person class.

(c) A wrong prediction: The model mistook the flower with potted plants.

(d) However, the model correctly identified the TV.

Figure 4: Correct classifications are depicted with blue support evidence bars, incorrect ones with red. The model trained on Pascal VOC with a DenseNet121 backbone barely classified the test sample in (a) as a cat, because all support images suggest that the model learned to look for two cat eyes. However, the test sample only contains one eye of the cat, and its support contributions suggest that the model is susceptible to detecting two eyes. Computing the contributions for the person class in (b) shows how the model utilizes the other portion of the input features. This shift is also present in the example (c) and (d).

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

## A APPENDIX

### A.1 ON THE FAITHFULNESS OF B-COS EXPLANATIONS

B-cos modles are piece-wise linear models that allows to summarize each input as an input-dependent linear transform $f(x) = \mathbf{W}_{1 \to L}(x)x$, derived in Eq. 3. Explanations are computed in terms of the weight matrix $\mathbf{W}_{1 \to L}(x)$ by multiplying each feature with its matrix weight $\phi(x)_i = [\mathbf{W}_{1 \to L}]_j^T \odot x]_{(ch,i,j)}$. To show that computing explanations for a forward pass is faithful (i.e., satisfies below six axioms introduced by Sundararajan et al. (2017)), we need to show that they hold considering the transformation matrix $\mathbf{W}_{1 \to L}(x)$. While $\mathbf{W}_{1 \to L}(x)$ is indeed input-dependent, it is effectively fixed during the computation of the explanation for its specific input $x$. This is because the explanations $\phi(f, X)$ aim to attribute the output $f(x)$ to the input features of $x$ using the weights at that point. Therefore, we assume a fixed $\mathbf{W}_{1 \to L}(x)$ when explaining the forward pass of an input $x$, and acknowledge that while $\mathbf{W}_{1 \to L}(x)$ varies across different inputs, it remains constant within the context of computing local explanations $\phi(f, x)$ for a particular $x$. In addition, we consider the case of single-class prediction, where the weight matrix simplifies to a single-column form ($\mathbf{W}_{1 \to L}(X) \in \mathbb{R}$) for in input vector $X \in \mathbb{R}$. This allows us to focus on the core properties of the explanation mechanism without loss of generality since explaining the prediction of a class in the multiclass setting is analogous to explaining its corresponding column vector of the matrix $\mathbf{W}_{1 \to L}(x)$.

We abbreviate $\mathbf{W}_{1 \to L}(x)$ with $W(X)$ for ease of notation in the following. Let WASUP be defined as $f(X) = W(X) \cdot X$ and its contributions similarly by $\phi(f, X) = W(X) \odot X = (W(X)_1 \times X_1, \ldots, W(X)_N \times X_N)$ (remark that $\cdot$ denotes the scalar product and $\odot$ the element-wise multiplication, $\times$ the multiplication of real numbers).

Now, we reformulate $f(X)$:

$$f(X) = W(X) \cdot X = \sum_{i=1}^{N} W(X)_i \times X_i = \sum_{i=1}^{N} \phi(f, X)_i. \tag{8}$$

**Completeness:** *The sum of feature attributions should add up to the model output.*

This follows trivially from the reformulation in Eq.8.

**Sensitivity:** *If changing only one feature's value changes the prediction of the model, this feature's attribution should be non-zero.*

We reformulate: $f(X) = \sum_{j=1}^{N} \phi(f, X)_j = \phi(f, X)_i + \sum_{j \neq i}^{N} \phi(f, X)_j$.

Modifying $X$ only in $i$, denoted by $\hat{X}$, and $f(X) \neq f(\hat{X})$ yields:

$f(X) = \phi(f, X)_i + \sum_{j \neq i}^{N} \phi(f, X)_j \neq f(\hat{X}) = \phi(f, \hat{X})_i + \sum_{j \neq i}^{N} \phi(f, X)_j$.

and allows to substitute $\sum_{j \neq i}^{N} \phi(f, X)_j$.

$\implies \phi(f, X)_i \neq \phi(f, \hat{X})_i$, which means that at least one of the two attributions in non-zero.

**Implementation Invariance** *If two models are functionally equivalent, i.e. the outputs are equal for all inputs, despite having different implementations, their contributions should always be identical.*

Let $f_1, f_2$ be implementations.

Then: $f_1 = W_1(X) \cdot X, f_2 = W_2(X) \cdot X, \forall X \implies W_1(X) = W_2(X), \forall X$.

It follows that $W_1(X) \odot X = W_2(X) \odot X, \forall X \implies \phi(f_1, X) = \phi(f_2, X), \forall X$.

**Dummy** *If a model $f$ does not depend on some feature $X_i$, its attribution should always be zero.*

If $f(X)$ is does not depend on some feature $X_i$, it follows that $W(X)_i \times X_i = 0, \forall X_i \in \mathbb{R}$

$\implies W(X)_i = 0 \, \forall, X_i \in \mathbb{R}$

$$\implies \phi(f, X)_i = 0, \ \forall X_i \in \mathbb{R}.$$

**Linearity**  *If the output of a model is a linear combination of two models, the attribution of the combined model should be the weighted sum of the contributions of the original models.*

We need to show that $f(X) = \alpha f_1(X) + \beta f_2(X) \implies \phi(f, X) = \alpha \phi(f_1, X) + \beta \phi(f_2, X)$.

$f(X) = \alpha f_1(X) + \beta f_2(X) = \alpha(\sum_{i=1}^{N} W_1(X)_i \times X_i) + \beta(\sum_{i=1}^{N} W_2(X)_i \times X_i) = \alpha(\sum_{i=1}^{N} \phi(f_1, X)_i) + \beta(\sum_{i=1}^{N} \phi(f_2, X)_i) = \alpha \phi(f_1, X) + \beta \phi(f_2, X)$.

**Symmetry-Preserving**  *If swapping two features does not change the model output for all possible values, they should have identical attributions.*

Let $X_i, X_j$ be two features.

If swapping the two does not change the model output for all possible values, it follows that

$W(X)_i \times X_i = W(X)_j \times X_j, \ \forall X_i, X_j \in \mathbb{R}$

$\implies \phi(f, X)_i = \phi(f, X)_j, \ \forall X_i, X_j \in \mathbb{R}.$

### A.2 ON THE FAITHFULNESS OF SUPPORT VECTORS

On a theoretical level, the global explanations are the explanations of an intermediate B-cos neuron. Thus, they satisfy the above axioms as well and can be interpreted as the information encoded in the respective support vector.

When providing local explanations, we view support vectors as the weights of a B-cos linear transform. Hence, computing the explanation of the prediction of the class in terms of the output yields the input features compressed in any of the support vectors; and the log-probability scores of each support vector indicate to which extend each was found.

**Justification for ReLU in WASUP**  In contrast to other functions that could be used to implement $\oplus$, the ReLU activation does not alter the range of the input to potentially huge or tiny numbers like, e.g. the exponential. However, ReLU activations are commonly scrutinized for their role in setting negative activations to zero, which can cause deep neural networks to interpret this suppression as the absence of a feature. Thus, many XAI methods (e.g. gradient-based) yield misleading attribution maps, as they cannot account for this type of contribution. However, this phenomenon is tightly controlled in WASUP, where the sole operation following the ReLU activation is a similarity computation between the latent vectors of two images. As a result, the absence of a feature does not, by design, affect the probability of class logits, rendering it valid to disregard any input and its explanation from neurons corresponding to such features.

### A.3 IMPLEMENTATION AND TRAINING DETAILS

**Optimization**  All models were optimized using the AdamW optimizer with fixed hyperparameters: weight decay set to 0.0, betas configured to (0.9, 0.999), and an epsilon value of 1e-08. We conducted an extensive grid search over the hyperparameter space (see Tab. A.2) to identify the optimal settings based on accuracy. The training employed a custom learning rate scheduling policy, which began with a linear warm-up phase over the first two epochs, increasing the learning rate from 10% to the initial learning rate. This was followed by maintaining the initial learning rate until 50% of the total training iterations were completed. Subsequently, the learning rate was decayed by a factor of 0.5 at every subsequent 10% milestone of the total iterations. Batch sizes were set to 32 for all datasets except RSNA, which utilized a batch size of 64. The number of training epochs was standardized to 50 across all datasets, with RSNA trained for 100 epochs. Pretrained feature extractor weights were sourced from torchvision `https://pytorch.org/vision/stable/index.html` and the official B-cos repository `https://github.com/B-cos/B-cos-v2` for all experiments except those involving RSNA.

Model-specific parameters included setting B-cos $B = 2$ and using three support samples. For WASUP, the grid search explored learning rates of 0.01, 0.003, 0.001, and 0.0003 for the RSNA

dataset, and 0.0022, 0.001, 0.00022, 0.0001, 0.000022, and 0.00001 for other datasets. Temperature parameters tested were 10, 30, and 50 for RSNA, and 10 and 30 for the remaining datasets. For BlackBox models, the grid search included the same learning rates as WASUPand additionally varied the weight decay parameter, evaluating values of 0.0 and 0.0001. The best-performing models were selected based on their accuracy metrics.

Table A.2: Best configurations and performance of each model. WASUP consistently performs marginally better ($\geq +0.29\%$) than the black box with the same backbone on Pascal VOC. On Stanford Dogs, the black box outperforms WASUP by $3.7\%$. On RSNA, the difference ($0.65\%$) marginally favors the black box, although a repeatability study with different random weight initializations is needed to confirm this finding (note that models were only randomly initialized in the RSNA experiment).

| Method | Model | Best LR | Best WD | Best Temp. | Acc (%) |
|---|---|---|---|---|---|
| BlackBox | VOC - DenseNet121 | $2.2 \times 10^{-5}$ | 0.0 | - | 96.42 |
| | VOC - ResNet50 | $1.0 \times 10^{-5}$ | 0.0 | - | 96.71 |
| | VOC - VitC | $1.0 \times 10^{-4}$ | 0.0 | - | 96.35 |
| | StanfordDogs - ResNet50 | $1.0 \times 10^{-5}$ | 0.0 | - | 87.16 |
| | RSNA - DenseNet121 | $3.0 \times 10^{-4}$ | $1.0 \times 10^{-4}$ | - | 79.78 |
| WASUP | VOC - DenseNet121 | $2.2 \times 10^{-4}$ | - | 30 | 96.73 |
| | VOC - ResNet50 | $1.0 \times 10^{-4}$ | - | 30 | 97.00 |
| | VOC - VitC | $2.2 \times 10^{-5}$ | - | 10 | 96.72 |
| | StanfordDogs - ResNet50 | $1.0 \times 10^{-4}$ | - | 10 | 83.46 |
| | RSNA - DenseNet121 | $3.0 \times 10^{-4}$ | - | 50 | 79.13 |

**Architecture** All backbones are extended by a linear projection layer to extract feature vectors $\in \mathbb{R}^d = 128$. The $\oplus$ function is implemented as a ReLU activation in the evidence predictor $\mathcal{E}$. In multi-label classification settings, only training samples with a single class label are considered for support labels.

**Datasets, Data Pre-Processing and Augmentation** In our study, we adhered to standard dataset splits to ensure consistency and comparability with existing research. Specifically, for Stanford Dogs, and Pascal VOC, we utilized the official training and testing partitions as provided. Regarding the RSNA dataset, we employed the training set from Stage 2 of the RSNA challenge and further subdivided it by randomly sampling 25% of the training data to form a separate test set, ensuring that the splits were stratified by class label. Additionally, we verified that the distributions across sex remained consistent following the splitting process, as illustrated in Tab. A.3.

Table A.3: Dataset statistics of the RSNA dataset.

| Dataset Split | Patient Sex | Target | Number of Samples |
|---|---|---|---|
| Training Set | F | No Opacity | 6,770 |
| | F | Opacity | 1,873 |
| | M | No Opacity | 8,734 |
| | M | Opacity | 2,636 |
| Test Set | F | No Opacity | 2,246 |
| | F | Opacity | 629 |
| | M | No Opacity | 2,922 |
| | M | Opacity | 874 |

For non-Bcos models, we applied standard normalization using the mean and standard deviation calculated from the training set. Specifically, images were standardized with mean values $[0.485, 0.456, 0.406]$ and standard deviations $[0.229, 0.224, 0.225]$.

Data pre-processing for the Pascal VOC and Stanford Dogs datasets included a `RandomResizedCrop` of torchvision to a target size of 224 pixels using bilinear interpolation.

We implemented a `RandomHorizontalFlip` with a probability of 0.5. For non-Bcos models, standardization was performed as mentioned above. Additionally, we employed `RandomErasing` with a probability of 0.5 and a scale range of 2% to 33% of the image area.

For the RSNA dataset, intensity rescaling was conducted to map the maximum Hounsfield Units to a range between 0 and 1. Data augmentation techniques included a `RandomHorizontalFlip` with a probability of 0.5 and a `RandomAffine` transformation. The `RandomAffine` parameters consisted of rotations up to 45 degrees, translations of $\pm15\%$ in each direction, and scaling factors ranging from 0.85 to 1.15. We also applied `RandomErasing` with a probability of 0.5, using patch sizes between 5% and 20% of the image area.

## A.4 ADDITIONAL FIGURES

### A.4.1 VOC - RESNET50

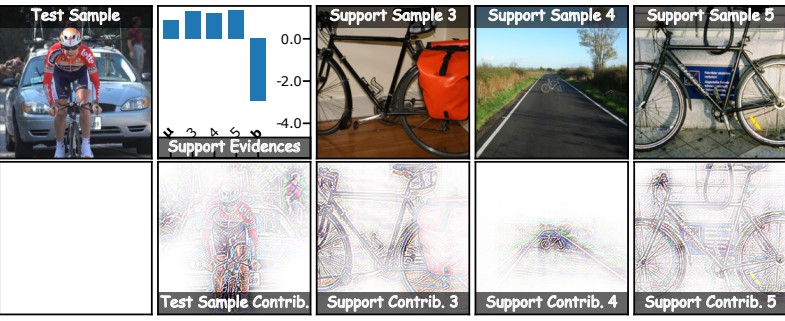

(a) Prediction for the bicycle class.

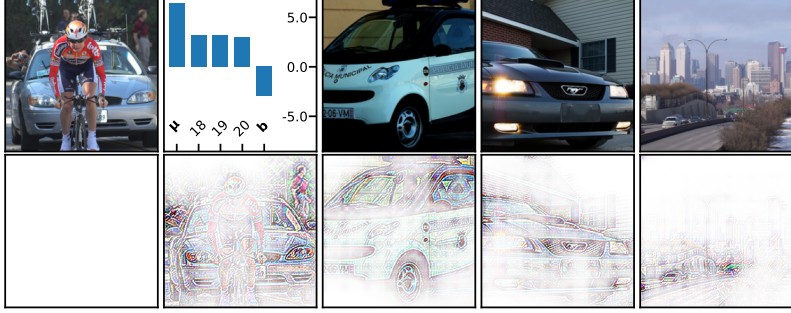

(b) Prediction of the car class. Note the shift in contributions compared to (a).

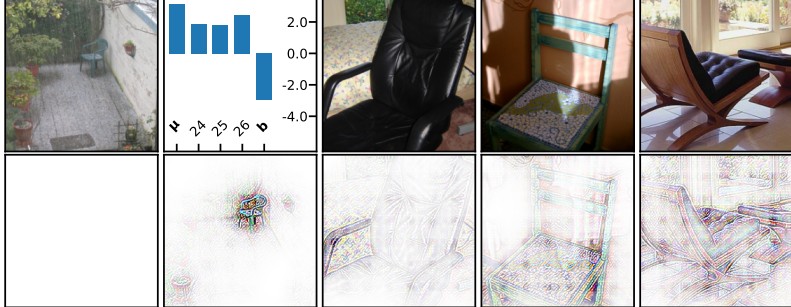

(c) The model correctly found the chair.

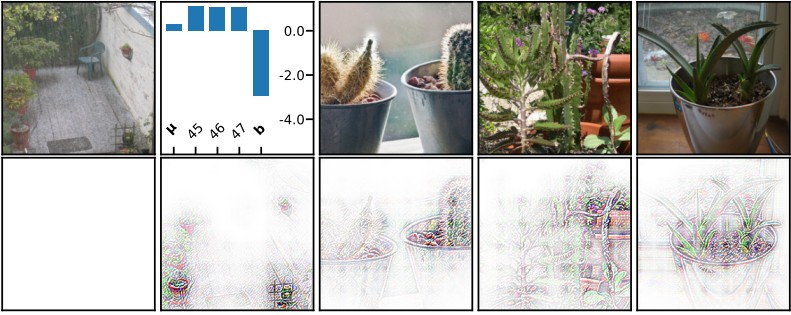

(d) And also the potted plants. Note the shift in contributions compared to (c).

Figure A.5: In this and the following figures, they layout introduces marginal changes compared to the main figure. Refer to (a) for the respective explanations. This model was trained with a ResNet50 backbone on VOC.

(a) Prediction for the chair class.

(b) Wrong prediction according to the label of the dining table.

(c) Wrong prediction according to the label of a person.

(d) And a wrong prediction, as the flower on the table was mistaken with a potted plant.

Figure A.6: VOC ResNet50

### A.4.2 VOC - VITC

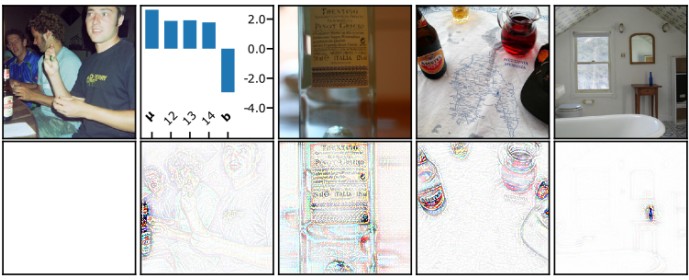

(a) Prediction for the bottle class.

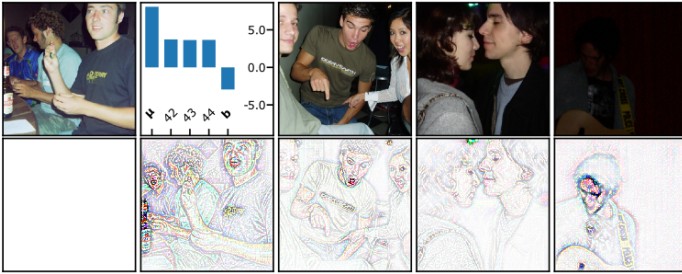

(b) Prediction of the person class.

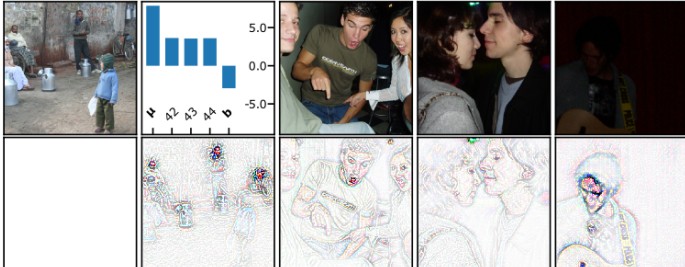

(c) For this test sample, this backbone was the only one not to mistake the milk jugs with bottles.

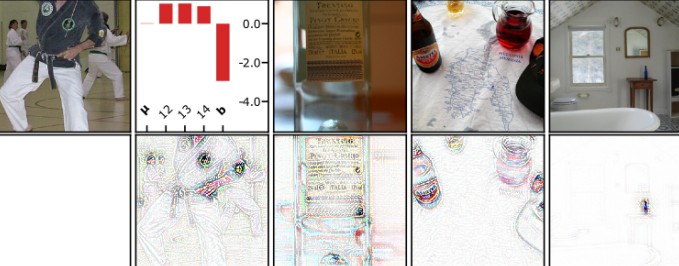

(d) Wrong prediction: The martial arts logo was mistaken with the labels of the first two support samples.

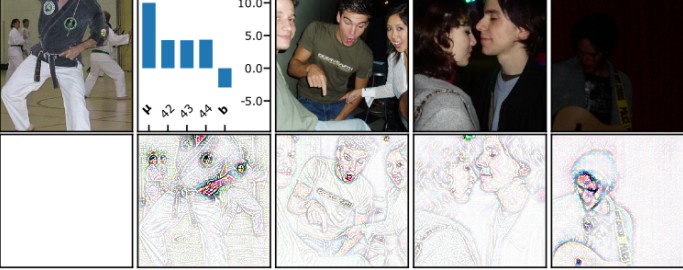

(e) Person was correctly classified.

Figure A.7: VOC VitC

### A.4.3  DOGS - RESNET50

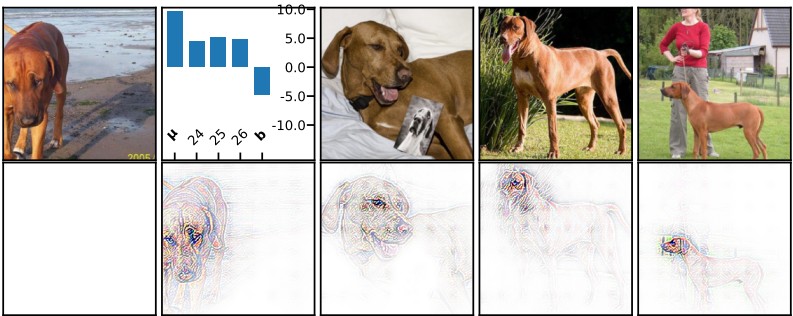

(a) Correct classification. Note the insensitivity to the background.

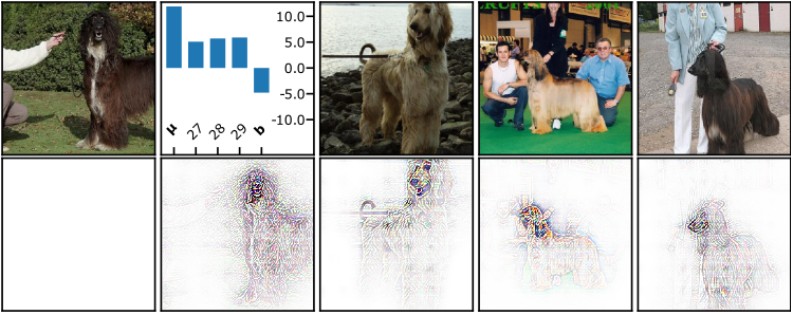

(b) Correct classification. Note the insensitivity to the background.

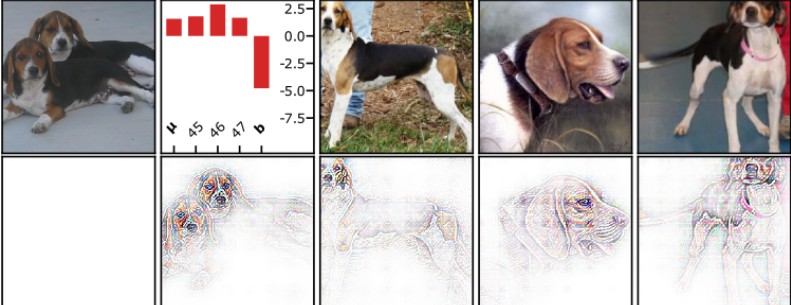

(c) Incorrect classification. The model was very unsure about the class...

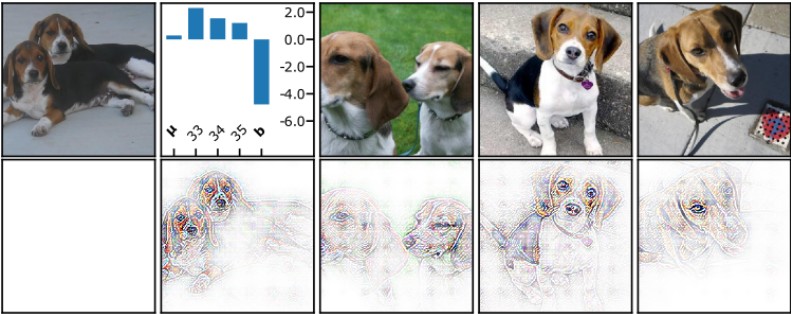

(d) The model found very little evidence for two of the three support samples.

Figure A.8: ...and the correst class probability suggests that the two breeds are indeed pretty similar.

(a) Correct classification. Note the insensitivity to the background.

(b) Correct classification. Note the insensitivity to the background.

(c) Incorrect classification. The explanations suggest that the dog on the left was prioritized by the network...

(d) ...in contrast to the (arguable) true label. Compared to support samples, the tip of the nose is relatively bright.

Figure A.9: Dogs Dataset ResNet50

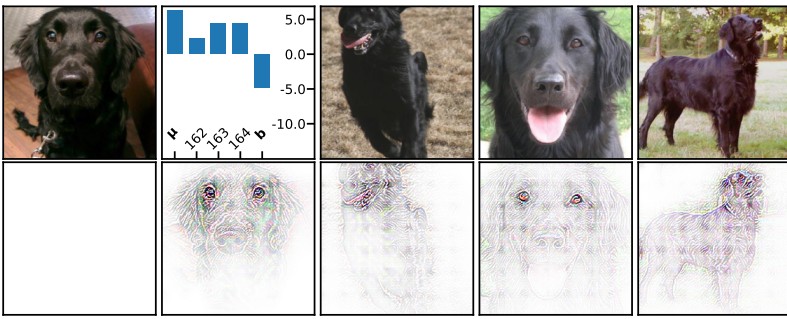

(a) Correct classification. Note how different poses of the support samples lead to different contributions, which is also evident in ...

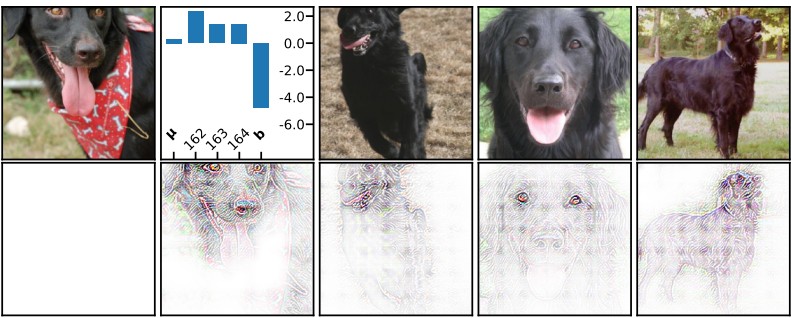

(b) ...this figure. Here, the first support sample is most similar to the test image, which is also cropped along the eyeline.

Figure A.10: Dogs Dataset ResNet50

### A.4.4 RSNA - DENSENET

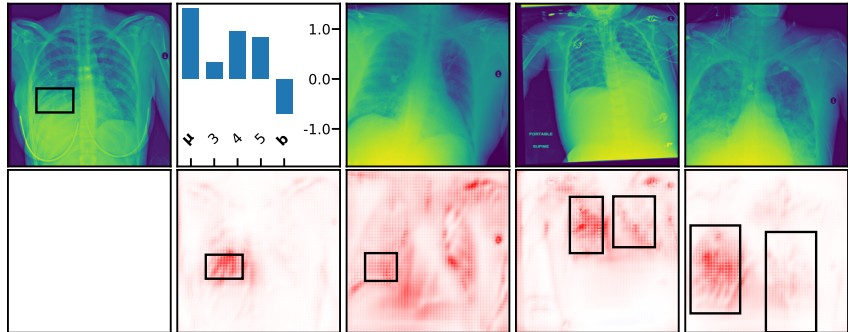

(a) Correct classification of an occlusion present sample.

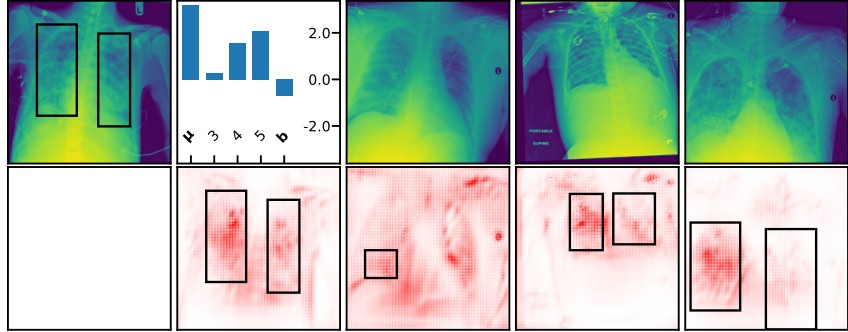

(b) Correct classification of an occlusion present sample.

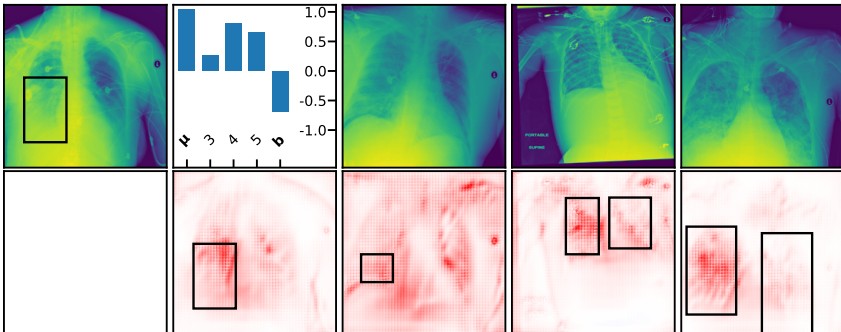

(c) Correct classification of occlusion present sample.

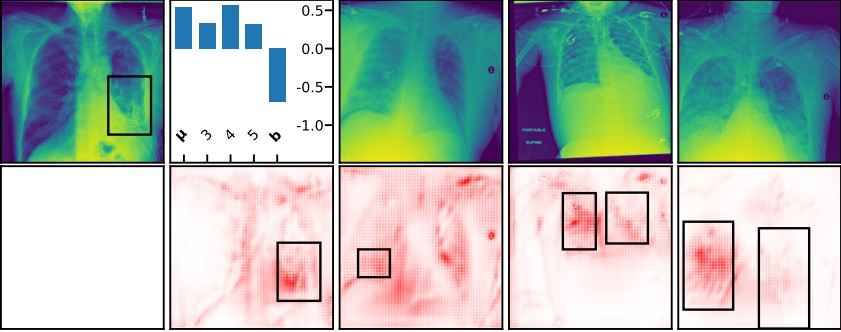

(d) Correct classification of an occlusion present sample.

Figure A.11

### A.4.5 VOC - DENSENET121 PROTOTYPES AND B-COS SIMILARITIES

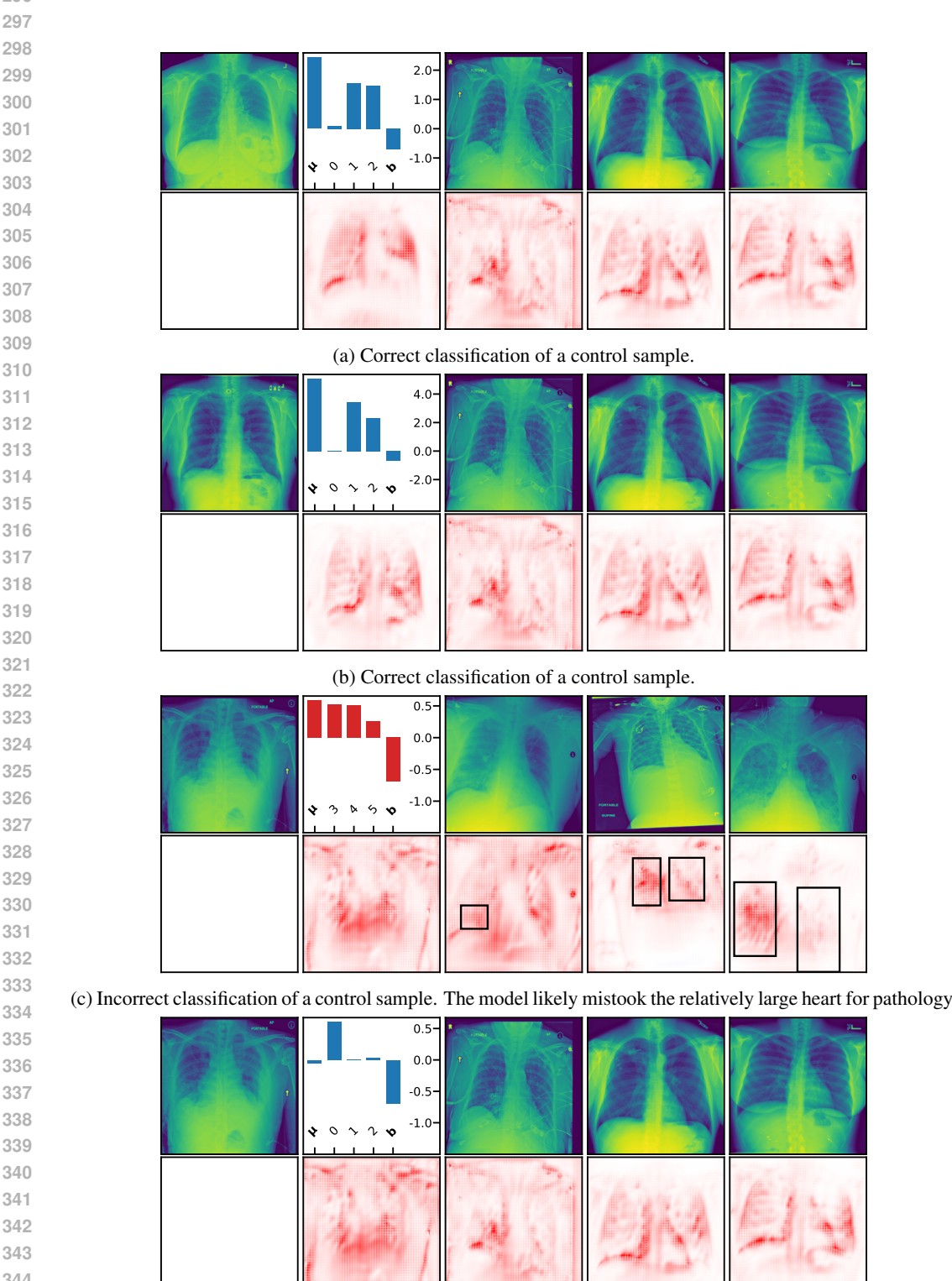

(a) Correct classification of a control sample.

(b) Correct classification of a control sample.

(c) Incorrect classification of a control sample. The model likely mistook the relatively large heart for pathology.

(d) The model found very little evidence for two of the three support samples.

Figure A.12

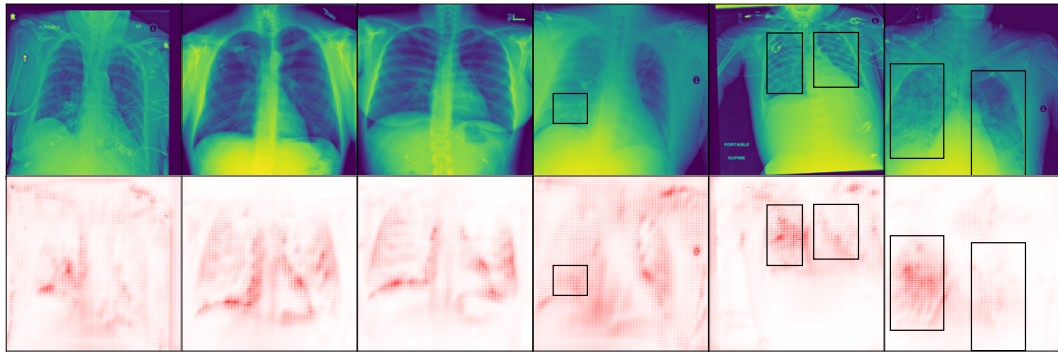

Figure A.13: Prototypes of WASUP trained with a DenseNet121 backbone on RSNA. Bounding boxes indicate where medical staff found occlusion of the lungs. The first three columns are the support samples of controls, the next three are support samples with occlusion present.

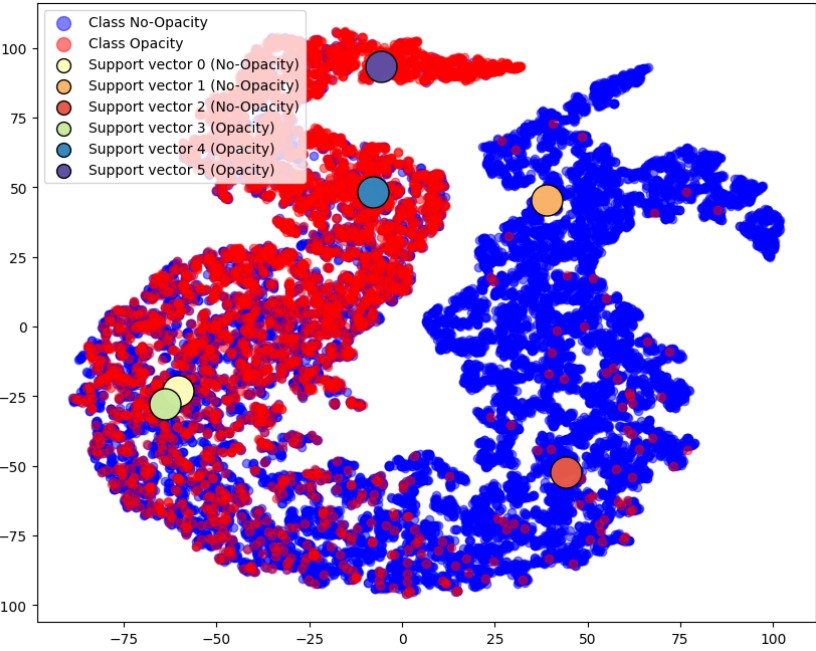

Figure A.14: t-SNE plot of the latent vectors of the RSNA training set. Red denotes samples of the class "opacity" and blue the class "no opacity". Support vector 0 and support vector 3 are very close in the projected latent visualization.

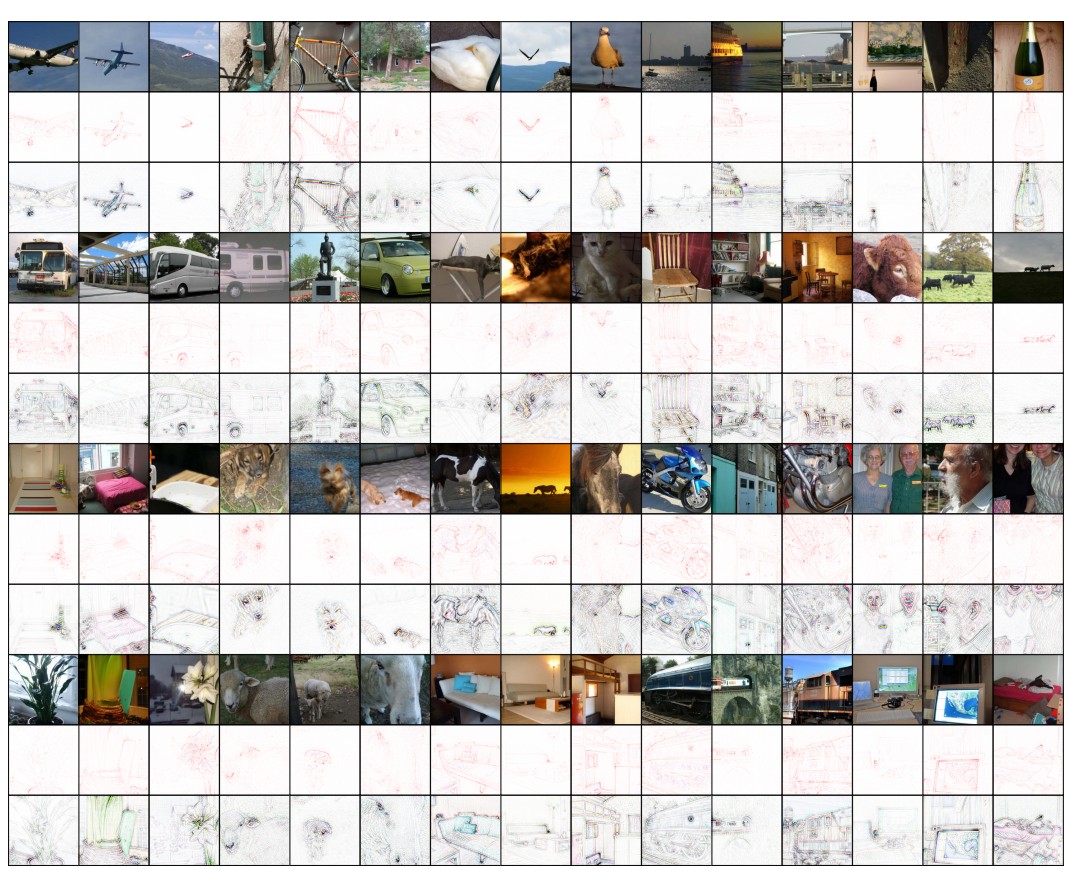

Figure A.15: Global explanations for WASUP trained on Pascal VOC with a DenseNet121 backbone. The respective second rows show the raw contribution maps $\phi^L_{j=c}$, in which red denotes positive contributions and blue (absent) negative contributions. The respective third columns present RGBA explanations.

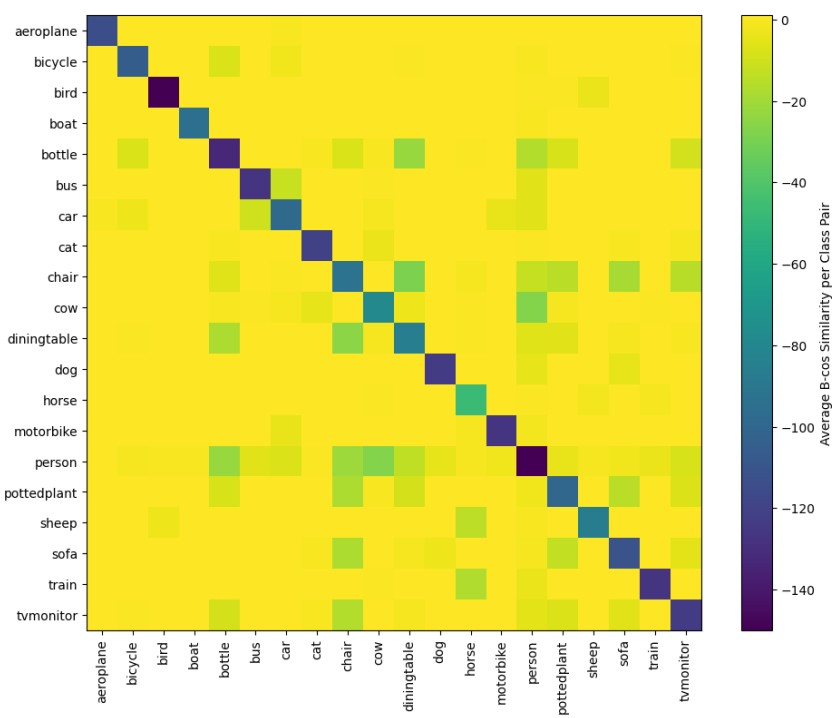

Figure A.16: Heatmap of Intra-class and Inter-class B-cos Similarities for WASUP trained on Pascal VOC with a DenseNet121 backbone.