# OpenReview forum: "WASUP: Interpretable Classification with Weight-Input Alignment and Class-Discriminative SUPports Vectors"
_ICLR.cc/2025/Conference — ICLR 2025 Conference Withdrawn Submission_

### Official Review · Reviewer_S5A2 · 2024-10-22

**Soundness:** 1
**Presentation:** 1
**Contribution:** 2
**Rating:** 1
**Confidence:** 5

**Summary:**

The paper describes an inherently interpretable model that consists of a B-cos network, which enables faithful localization in the input, and combines it with a Nadaraya-Watson head. Effectively, class training samples are saved as prototypical samples and during inference the class is predicted, which class prototypes are the most similar to the test image in b-cos feature space. As the similarity is measured in b-cos space, it can be faithfully mapped to the input space. The method is applied to single- and multi-label classification.

**Strengths:**

The paper follows a very promising goal of building locally and globally inherently interpretable models.

The paper effectively combines two current ideas in the field, b-cos networks and  Nadaraya-Watson heads.

The idea and method is well presented and easy to understand. They are supported by high quality clear figures 1-4.

The model provides faithful localization due to the b-cos network and generates global class explanations as collection of class examples.

**Weaknesses:**

The paper is significantly too long and can not be judged in 10 pages.
Only the appendix contains results on multiple datasets and the theoretical evaluations.
A significant part of the paper, pretty much the entire results section, are not understandable without looking at the appendix.
Therefore, the presentation and soundness are "poor". If the paper length was 16 pages, they would be "fair" soundness and "good" presentation.

The novelty is fairly limited, as it combines B-cos networks with prototypical methods, for which the sota methods are backbone-independent. Specifically, (1) with b-cos backbone would be fairly similar.

While there is a section for it (l. 311), I am not sure what the global explanation looks like.

The writing could be polished in these lines:
118, 190, 315 (unclear), 467

Using support-features and training the model that way most likely causes uninterpretable not human-like similarities to emerge, as measured in the Hoffman et al. paper cited, or dicussed in (2), since the features are already encoding the classification. This is also evident in e.g. 4.c, in which plant soil is similar to the flower, or in the spurious correlation of the keyboard in 4d. A human study as in the Hoffman paper would be beneficial to test if a human would be able to predict the similarities.

As noted by the authors in the appendix, a more thorough investigation of  the result with random seeds would be beneficial.

The evaluation is missing the default datasets for interpretable image classification, CUB and StanfordCars, and a comparison to competitors, such as the vanilla b-cos model, or other interpretable models with / without b-cos backbone such as PIP-Net, ProtoPool(1) or Q-SENN(2).

(1) Rymarczyk, Dawid, et al. "Interpretable image classification with differentiable prototypes assignment." European Conference on Computer Vision. Cham: Springer Nature Switzerland, 2022.
(2) Norrenbrock, Thomas, Marco Rudolph, and Bodo Rosenhahn. "Q-senn: Quantized self-explaining neural networks." Proceedings of the AAAI Conference on Artificial Intelligence. Vol. 38. No. 19. 2024.

**Questions:**

How many class prototypes exist? Just 3? What is the impact of that parameter?

How does this method compare to other SOTA interpretable methods with b-cos backbones?
Is there a fundamental benefit of applying the NW-Head to b-cos compared to other approaches for class features / prototypes?

Can the feature encoder be trained unsupervised, to ensure that image features are not encoding a  classifier decision?

Can the class support vectors be cropped according to the saliency map during training, to ensure that e.g. Fig A.5a, Support sample 4 does not respond to grass?

---

### Official Review · Reviewer_mcS9 · 2024-10-31

**Soundness:** 2
**Presentation:** 3
**Contribution:** 2
**Rating:** 5
**Confidence:** 4

**Summary:**

The authors propose an inherently interpretable network that can implement any neural network architecture and provide local, global and faithful explanations. The idea is to use support vectors from a class to evaluate their similarity to an input image embedding using the Nadaraya-Watson head and visualize pixel importance using a B-cos network structure.

**Strengths:**

→ Paper structure and writing: The paper is well written and easy to follow. I especially appreciate the theoretical basis of the methodology explained, including the B-Cos networks and the Nadaraya-Watson head method.

→Faithful explanations: There is a need to provide faithful explanations and not just approximations of the behavior of the model. I believe that the authors had good insights in proposing a network that preserves the axioms defined by Sundararajan.

→Visual and Quantitative Explanations: The paper provides pixel-wise visual explanations as well as quantitative evidence of the support vectors that influence the class decision. This kind of explanation can be well used when searching for model errors.

**Weaknesses:**

→Images used to construct support vectors: It is not clear to me how you select the initial images that generate the initial support vectors. I believe that randomly selecting images in the beginning from a class could lead to a lack of variability if they are not well sampled, and this could affect the learning process. What happens if all the images selected from the dataset are similar and provide a single type of support vector, or what happens if the dataset used to sample the images is biased? I think it would be interesting to see results from different initially selected subsets of the dataset to construct these vectors. I would also suggest using a biased dataset to show what happens when we sample from a biased training set. Can we see the bias in the resulting prototype images?

You might also include some experiments on: Number of images to construct the support vectors; how the support vectors change when you change the set of images; what each support vector represents (semantically).

Also, during the training process you mention that you use k-means to extract the centroids, it would be interesting to see experiments with k-means parameters. I imagine the exact number of classes as k is not enough to describe all the possible differences within each class. What was the best k in this process?

→  Global explanations: as you mentioned, if nothing is similar to a sample, it will show the most similar support vector (even if it is not similar) with smaller contribution. But what happens if the image is far from all the support vectors? Can we say that the model does not know it or that we did not construct the support vectors correctly?

As I understood, we could analyze the output space of the model to have the proximity on an input image to others in the dataset. But how do you analyze the global behavior of the model? Is it correct or biased? It would be interesting to see some examples of global explanations (not just sample-based explanations).

→ I don't understand exactly why you don't use the original vector representation and only the positive-valued vectors. What happens without the relu? Also, why can we not have negative attributions? Doesn't that mean that this feature is not present in the class according to the model's knowledge? It would be interesting to extend the explanation about this.

→ Interpretation: I can see that the visualization focuses attention on the class object, but I cannot be sure with this visualization which features in the object are important and what differentiates one support vector from the other in the same class. To me, it is a good explanation, but I find it difficult to interpret.

→ Comparison with other methods:  I would suggest that the authors also compare their methodology (even with a qualitative analysis) with methods such as ACE that also provide global explanations.

**Questions:**

Some questions were previously presented in the weaknesses section.

Other questions:

→ I understand that if the network is linear, the found contributions of the input features are faithful. However, how can you determine to which class support vector each part of the test sample contributions are aligned? In other words, can you determine specific image regions of the contributions? You may have multiple support vectors per class, so how to decompose this knowledge?

→ I would also like to see different configurations and their performance, is it difficult to tune the training to get the reported accuracies?

→ Is it possible to adapt this methodology to textual networks?

**Details Of Ethics Concerns:**

No concerns.

---

### Official Review · Reviewer_igfe · 2024-11-02

**Soundness:** 2
**Presentation:** 2
**Contribution:** 3
**Rating:** 3
**Confidence:** 4

**Summary:**

The paper proposes to combine B-cos networks as feature extractor and a relatively common version of few-shot learning. The authors evaluate the resulting model on three datasets and provide exemplary attribution maps. They show that the proposed method satisfies axioms from the Integrated Gradients paper (Sundararajan, 2017).

**Strengths:**

They fuse few-shot learning with the powerful B-cos networks. They perform experiments on 3 data sets. Nice attribution maps!

**Weaknesses:**

The weaknesses in short are:

- Masking limitations in novelty.
- A weak measurable evaluation with missing ablation study.
- not measuring faithfulness and reliance on a purely axiomatic definition of it.

- Masking limitations in novelty:
lets be clear:

The Nadaraya-Watson head is (a simplified version of) few-shot learning. It is a softmax over negative distances between support samples and the test image.

While it is appreciated that Wang and Sabuncu, 2022, gave a name to their analysis in order to emphasize a predecessor of few-shot learning, using this term in this paper suggests a larger or different novelty than there actually is.
It should be made prominently clear in the manuscript that the Nadaraya-Watson head is effectively few-shot learning (seemingly without sampling random subsets of classes).

The evidence head is a standard few shot head. Taking the positive part is a ReLU applied on a feature map. Again, that is renaming common parts to sound uncommon / novel.

Masking limitations in novelty in such a way is disliked by the reviewer. This results in a low score for presentation.

- A weak measurable evaluation with missing ablation study.

Table A.2 WASUP is not compared against pure B-cos backbones on which it builds on, but only with a black-box standard CNN.

By that one cannot distinguish whether the contributions are actually mostly from the B-cos network or whether the few-shot head plays any role in (a) predictive performance or (b) attribution map quality.

An ablation study is missing to quantify how much of the predictive performance and attribution map quality comes from the B-cos network or the few-shot head. In the worst case the B-cos network alone does all the heavy lifting.

If one checks the B-cos networks, then it appears to be well possible that the attribution map quality originates predominantly from the B-cos networks (e.g Fig 3 in https://arxiv.org/pdf/2306.10898, the pointing game shows high spatial selectivity, but also Fig. 4).

--  not measuring faithfulness and reliance on a purely axiomatic definition of it.

"We prove that explanations provided by WASUP fulfill the axioms required to be faithful."

"To thoroughly assess the interpretability of WASUP, we conduct a theo-
retical evaluation of its explanations based on the axioms defined by Sundararajan et al. (2017), ..."

Faithfulness should be measurably quantified. That is a purely theoretical assessment.

Axioms are not a generally accepted definition of faithfulness. Axioms in this field are not uniquely determined. One can define different, non-compatible sets without the ability to rank or to exclude them.

For example, integrated gradients satisfy the axioms in their own paper, yet they have a ratther low measurable faithfulness under most data-driven faithfulness measures.


Minor:
1 . "The prediction of our model is a linear transform with the temperature and the bias term"
That is not true if stated like that, because the weights are scaled depending on the test sample. It is not a big issue though


2. Pascal VOC: as most classes have very few positives, the standard of evaluation is average precision, not accuracy. One should report average precision, not accuracy for this data set.

**Questions:**

How much is the accuracy (2 of the 3 datasets)/average precision (for VOC) for the network which uses B-cos networks but not few-shot head ?
If this experiment would be done, it would make the paper clearly more valuable, also with respect to the final rating.

How does numerical faithfulness evaluate for the black-box network / B-cos networks without few-shot head / WASUP ?

Not necessary, but of interest: if one removes the relu in the "evidence head", how much would that impact accuracy/mAP ?

Can you please make it clear in the manuscript that you are fusing B-cos feature extractors with few-shot learning ?

---

### Official Review · Reviewer_bGsc · 2024-11-03

**Soundness:** 2
**Presentation:** 2
**Contribution:** 2
**Rating:** 3
**Confidence:** 3

**Summary:**

In this paper, the authors proposed WASUP, an inherently interpretable neural network for image classification. In particular, a WASUP model combines a B-cos network with a classification head that learns a set of support vectors and classifies an input image based on its similarity with the support vectors in the latent space. During training, a WASUP model takes a trained B-cos network (trained using binary cross entropy loss) and learns a set of k support vectors for each class by applying k-means to the B-cos-extracted features of all training images belonging to that class, and then replacing the cluster centers with the closest training image features. The authors evaluated their WASUP models using PASCAL VOC (multi-label classification), Stanford Dogs, and RSNA, and found that their models achieved similar accuracy compared to the baseline (black-box) model while providing both local and global interpretability.

**Strengths:**

- Originality: The proposed method could be seen as an extension of the B-cos network.
- Quality: The proposed method generally makes sense.
- Clarity: Most of the paper is easy to follow.
- Significance: Interpretability is an important topic in machine learning.

**Weaknesses:**

- Originality: The paper is not novel. The support vectors are, in essence, prototypes (as in a ProtoPNet). The comparison between an input image's latent representations and support vectors is also similar to the comparison between the latent representations and prototypes in a ProtoPNet. The proposed method is simply a combination of a B-cos network and a ProtoPNet.
- Quality: The authors only compared the accuracy of their WASUP models with non-interpretable models. They did not compare with other interpretable models.
- Quality: There is no quantitative and qualitative comparison of interpretability with other interpretable models as well.
- Significance: Since the main ideas behind the paper are mostly explored in prior work and there is no novelty, the paper lacks significance.

**Questions:**

Since the binary cross entropy loss is used for training, could the classifier predict more than one class for a given test image, when there is only one ground-truth class (e.g., as in Stanford Dogs)?

**Details Of Ethics Concerns:**

N/A.

---

### Official Review · Reviewer_QZvc · 2024-11-04

**Soundness:** 3
**Presentation:** 2
**Contribution:** 2
**Rating:** 3
**Confidence:** 3

**Summary:**

The paper proposed an inherently interpretable neural network that allows case-by-case reasoning and provides faithful local & global explanations by combining the Nadaraya-Watson head with the B-cos techniques. Overall, the idea is intuitive and interesting.

**Strengths:**

1. The paper is written in easy-to-understand English.
2. A solid proof of faithfulness is provided.
3. The proposed method works on both CNN-based and ViT-based backbones. It successfully applies to general image classification tasks and the medical domain.

**Weaknesses:**

1. The paper’s contribution is limited. The faithfulness of the proposed model relies heavily on existing B-cos networks, while its global interpretability comes from a Nadaraya-Watson head. The work largely made engineering efforts to combine these two established ideas without introducing new insights.
2. The original design element in this paper—using k-means clustering to extract class-specific centroids for support vectors—is intuitive but lacks novelty.
3. The paper would benefit from more thorough proofreading by the authors, as numerous notational issues affect comprehension. For instance, the notation ||w||=1 on line 190 is factually incorrect, Equation 2 for \hat{W} is in recursive form but lacks an explanation of initial values, the term c' in lines 163–167 is undefined, etc.
4. The methodology is similar to prototype learning, and a stronger case for the interpretability of the approach could be made by including both numerical and visual comparisons.

**Questions:**

1. According to the original B-cos paper, they "observed an increase in training and inference time by up to 60% in comparison to baseline models of the same size." Does WASUP encounter similar or even greater computational overhead?
2. Is the numerical results reported in the paper statistically significant? It seems the performance change compared to the black-box models on Pascal VOC and RSNA is subtle. Could the authors clarify whether these changes are meaningful?

---

### Note · Authors · 2024-11-19

I have read and agree with the venue's withdrawal policy on behalf of myself and my co-authors.